# Flexible integration of free-standing nanowires into silicon photonics

Bigeng Chen[1], Hao Wu[1], Chenguang Xin[1], Daoxin Dai[1] & Limin Tong[1,2]

Silicon photonics has been developed successfully with a top-down fabrication technique to enable large-scale photonic integrated circuits with high reproducibility, but is limited intrinsically by the material capability for active or nonlinear applications. On the other hand, free-standing nanowires synthesized via a bottom-up growth present great material diversity and structural uniformity, but precisely assembling free-standing nanowires for on-demand photonic functionality remains a great challenge. Here we report hybrid integration of free-standing nanowires into silicon photonics with high flexibility by coupling free-standing nanowires onto target silicon waveguides that are simultaneously used for precise positioning. Coupling efficiency between a free-standing nanowire and a silicon waveguide is up to ~97% in the telecommunication band. A hybrid nonlinear-free-standing nanowires–silicon waveguides Mach–Zehnder interferometer and a racetrack resonator for significantly enhanced optical modulation are experimentally demonstrated, as well as hybrid active-free-standing nanowires–silicon waveguides circuits for light generation. These results suggest an alternative approach to flexible multifunctional on-chip nanophotonic devices.

---

[1] State Key Laboratory of Modern Optical Instrumentation, College of Optical Science and Engineering, Zhejiang University, Hangzhou 310027, China. [2] Collaborative Innovation Center of Extreme Optics, Shanxi University, Taiyuan 030006, China. Correspondence and requests for materials should be addressed to L.T. (email: phytong@zju.edu.cn)

Over the last decade, with a mature CMOS-compatible top-down technique, silicon photonics has been proved to be very promising for supercomputing and data-communications[1–6]. It features a variety of advantages including large-scale manufacturability, high reproducibility, and high integration density. Similar to the advancement of integrated electronics with the feature width of interconnect decreasing down to sub-10-nm level driven by Moore's law[7, 8], size miniaturization of integrated photonics also develops rapidly for higher operation speed, less power consumption, and so on[9–11]. However, compared to electric current in a metal wire with relatively low frequency, optical frequency signals in a waveguide are much more sensitive to the surface roughness, leading to roughness-induced optical loss in top-down fabrication structures[12]. For example, with RMS surface roughness down to 0.1-nm level[13], a subwavelength-diameter silica nanofiber can guide light with propagation losses well below 0.005 dB cm$^{-1}$ [14, 15], which is far beyond the reach of up-to-date fabrication techniques for silicon nanowaveguides (e.g., 0.4 dB cm$^{-1}$ for strip wave-guides[16] and 0.3 dB cm$^{-1}$ for ridge waveguides[17]). On the other hand, around the same time, relying on a bottom-up approach such as vapor–liquid–solid growth, the chemical and material communities have grown a variety of free-standing nanowires (FNWs) for electron and photon transportation[18–23]. These highly uniform one-dimensional nanostructures, with diameters of a few to hundreds of nanometers, show surface roughness down to atomic level (i.e., the same order of the silica nanofiber), offering an opportunity for deep-subwavelength optical wave-guiding with low optical loss[24, 25]. Moreover, benefiting from the bottom-up process that synthesizes nanowires from atomic-size nucleation sites, the nanoscale transverse cross sections of FNWs allow large lattice mismatch in single-crystalline structures[26–28], which offers much greater material diversity for active and nonlinear applications compared with the traditional top-down technique[29–34]. In order to merge the advantages of the bottom-up and the top-down techniques, previously there have been some successful studies on integrating FNWs with on-chip waveguides[35–37]. The big challenge is to assemble FNWs into functional photonic circuits with low coupling loss and high precision. In previous papers, the coupling efficiency is usually low (e.g., below 5%[34, 36]) due to the mismatch of mode overlapping and photon momentum.

Here we report flexible integration of FNWs and silicon photonics merging their advantages together via an elaborate near-field coupling structure. In our layout design, the as-fabricated on-chip waveguides provide precise positioning for the FNW assembling. As a result, the coupling efficiency between a FNW and a silicon waveguide (SW) can be up to ~97% in the wavelength range from 1,535 to 1,620 nm, corresponding to a minimum loss of ~0.14 dB. Hybrid photonic integrated circuits such as Mach–Zehnder interferometers (MZIs) and racetrack resonators are fabricated with nonlinear or active FNWs. Especially, using 405 nm switch pulses with a low peak power of 250 μW to pump on the CdS FNW arm of a hybrid MZI, we switch out signal pulses at 1.5 μm with a modulation depth (defined by the differential transmission $\Delta T/T$) of 100%, which is four times larger than that case when the SW arm is pumped. Besides, waveguided luminescence at telecommunication band is also realized by integrating active tellurite glass FNWs with silicon racetrack resonators.

## Results

**Optical near-field coupling between FNWs and SWs.** The integration of a FNW into a typical silicon photonic circuit is illustrated schematically in Fig. 1a, where a FNW is used to bridge the two SWs via side-by-side near-field optical coupling. On-chip SWs were fabricated with a silicon-on-insulator (SOI) wafer by the processes of E-beam lithography and subsequent dry-etching (see Methods section). An as-grown CdS FNW (Supplementary Note 1) was first transferred onto the silicon chip via

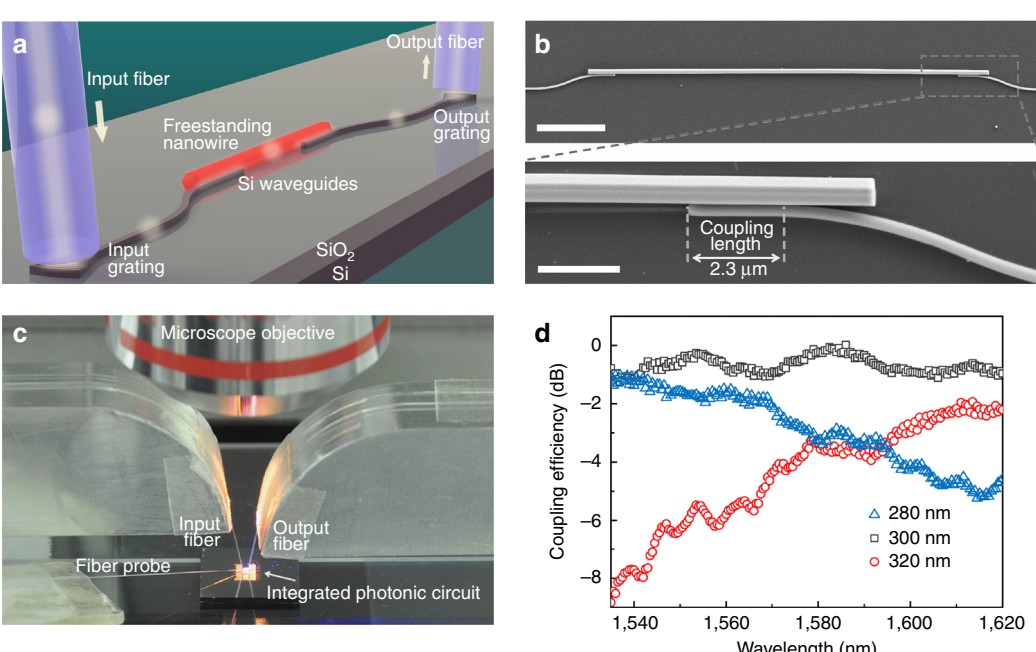

**Fig. 1** Integration of a FNW into a silicon chip using side-by-side near-field optical coupling. **a** Schematic illustration of the coupling scheme. The *white arrows* indicate the direction of light propagation. **b** *Upper panel:* scanning electron microscopy (SEM) images of a 730-nm-diameter CdS FNW coupled to 280-nm-width SWs at both ends. Scale bar, 10 μm. *Lower panel:* close-up image showing the pre-defined coupling length of about 2.3 μm. Scale bar, 2 μm. **c** Photograph of the measurement setup under an optical microscope. The silicon chip was illuminated by the microscope. **d** Measured coupling efficiency between a 860-nm-diameter CdS FNW and SWs with widths of 280, 300, and 320 nm, respectively. The coupling lengths are about 2.5 μm

micromanipulation, and then pushed to make close contacts with the two SWs at the ends by using a tapered fiber probe under an optical microscope (Fig. 1b and c, also see Supplementary Note 2). To achieve an efficient coupling, the diameter of the FNW was chosen to have phase-matching with the SW[38] (Supplementary Note 3) while the length of the coupling region was determined automatically by the layout of the SW (which consists of a straight section and an S-bend), as shown in Fig. 1b. Such a quasi-self-aligned assembling process is very helpful and robust for achieving the efficient coupling. For optical characterization, a broadband light at telecommunication band was coupled into/out from the chip via grating couplers. By measuring the transmissions of SW pairs (340-nm height, different widths) bridged by an identical 860-nm-diameter CdS FNW (Supplementary Note 5), we obtained the coupling efficiencies between the FNW and SWs with different core widths (Fig. 1d). Within the measured spectral range of 1,535–1,620 nm, the coupling efficiency of the FNW with a 300-nm-width waveguide is averagely very high, with a maximum value of about 97% around 1,585 nm after calibrating the propagation losses of the FNW and the SW (Supplementary Note 5). A small deviation to this optimal width (e.g., ± 20 nm) leads to notable wavelength-dependent efficiency, which is mainly due to the phase mismatches in these cases (Supplementary Notes 3 and 4).

**Integrated FNW–SW MZI for all-optical modulation**. The high-efficiency FNW-waveguide coupling scheme can be readily exploited to construct on-chip functional photonic devices. Figure 2a gives an SEM image of a hybrid MZI integrated with a CdS FNW connected with a U-shaped SW. Measured

transmission spectrum (Fig. 2b) shows evident interference fringes with extinction ratios up to 20 dB. The measured free spectral range (FSR) is about 3.1 nm around 1,598 nm, which agrees well with the calculated value of 3.0 nm. Small ripples in the spectrum are mainly owing to back reflection-induced Fabry–Perot resonance[39]. To generate free carriers for refractive index modulation similar to silicon optical modulators[9], here we illuminated chopped 405-nm-wavelength (3.06 eV, larger than the bandgaps of CdS and silicon) pulses (repetition rate ~67 Hz, pulse duration ~0.75 ms, and peak power ~250 μW) onto the CdS FNW with a spot size of about 6 μm$^2$ (area A in Fig. 2c), and measured the transmission response of a 1.5-μm CW signal light propagating along the MZI circuit (Fig. 2c, see also Supplementary Note 6). As shown in Fig. 2d, the signal light (*red line*) was modulated clearly by the switch light (*blue line*), suggesting a potential application of all-optical modulation[40]. The modulation depth was about 100%. For reference, when the SW (area B in Fig. 2c) was illuminated by the switch light under the same condition, the modulation was about four times weaker (*black line*). The results show that, compared with SW, CdS FNW presented higher carrier-induced optical nonlinearity, and thus offered higher modulation depth in the integrated circuit. It is worth to note that, benefiting from the material diversity[41], there are many other FNWs (e.g., KNbO$_3$[29]) can be used for various nonlinear photonic applications.

**Integrated FNW–SW resonator using a vertical coupling scheme**. Besides the side-by-side coupling scheme (Fig. 1a), we also bridge two SWs by placing a FNW in vertical contact, as illustrated in Fig. 3a. The measured coupling efficiency between a 770-nm-diameter CdS FNW and a 320-nm-width SW can go up to 87%

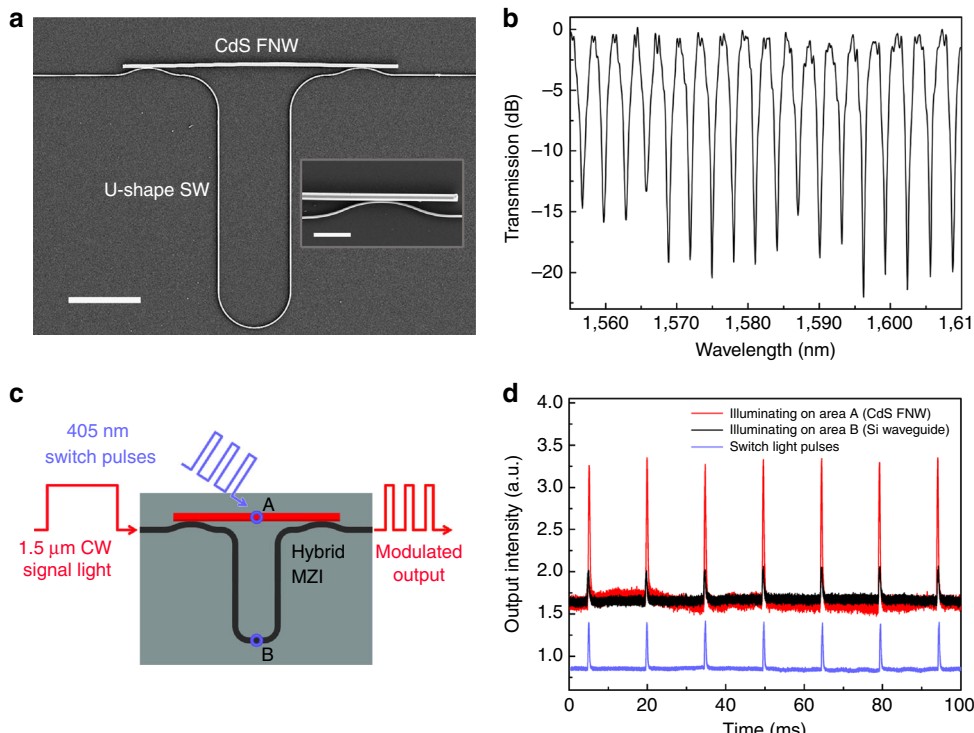

**Fig. 2** Integrated FNW–SW MZI for all-optical modulation. **a** SEM images of the MZI consisting of a U-shaped 300-nm-width SW and a 950-nm-diameter CdS FNW. The lengths of the SW arm and the FNW arm are about 183 and 59 μm, respectively. Scale bar, 20 μm. The inset shows a close-up view of the right-hand coupling region. Scale bar, 5 μm. **b** Transmission spectrum of the hybrid MZI. **c** Schematic diagram of all-optical modulation based on the hybrid MZI. *Blue circles* A and B indicate illuminating areas on the CdS FNW and the SW, respectively. **d** Modulated outputs from the MZI with 405-nm-wavelength switch pulses

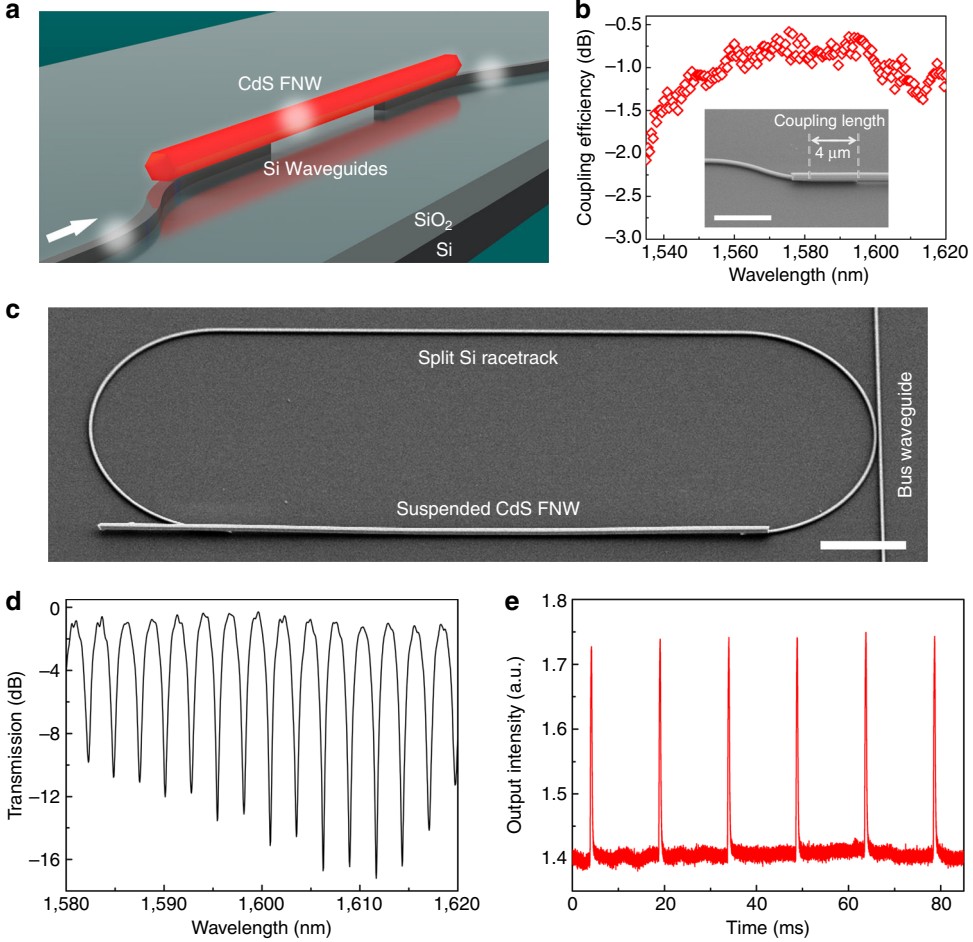

**Fig. 3** Integration of a FNW into a silicon chip using a vertical coupling scheme. **a** Schematic diagram of the vertical coupling structure suspending a FNW across two SWs collinearly. The *white arrow* indicates the direction of light propagation. **b** Wavelength-dependent coupling efficiency of a 770-nm-diameter CdS FNW with a 320-nm-width SW. Inset shows an SEM image of the coupling structure with a pre-defined coupling length of about 4 μm. Scale bar, 5 μm. **c** SEM image of a hybrid FNW–SW racetrack resonator. A bus waveguide is used for optical signal launching and collection. The overall length of the racetrack is consisted by a 60-μm CdS FNW and a 154-μm SW. Scale bar, 10 μm. **d** Transmission spectrum of the hybrid resonator. **e** Modulated signal output by all-optical modulation when the FNW is excited by 405-nm switch pulses

around 1575 nm (Fig. 3b). Using this vertical coupling approach, we constructed a hybrid resonator by bridging the gap of a racetrack waveguide with a 730-nm-diameter CdS FNW (Fig. 3c). Figure 3d gives the resonance spectrum of the fabricated hybrid resonator with extinction ratios up to 16 dB and quality factors of ~1,400. The measured FSR of 2.7 nm agrees well with the calculated value of 2.6 nm. By illuminating the CdS FNW with a 405-nm switch light, evident modulated output is clearly observed (Fig. 3e).

**Integrated FNW–SW resonators for on-chip light generation.** In additional to the nonlinear FNWs, we can also integrate rare-earth-doped FNWs with silicon photonics for on-chip light generation. For example, by vertically coupling an $Er^{3+}/Yb^{3+}$-codoped tellurite glass FNW to a silicon racetrack resonator (Fig. 4a), and exciting the FNW by a 976-nm CW laser (Fig. 4b), we obtained spectrum-sliced photoluminescence[42, 43] around the telecommunication band at the drop waveguide (Fig. 4c). Tellurite glass FNWs were directly drawn from doped bulk glasses as described in a previous study[44]. Green light observed on the excited FNW is attributed to the up-conversion luminescence of

$Er^{3+}$ ions[45]. The modal spacing of 4.0 nm around 1,540 nm agrees well with the calculated value of 3.9 nm. The spacing is determined by the perimeters of resonators. For example, we obtained a modal spacing of 10.8 nm by using a 50-μm-perimeter ring resonator (Fig. 4e).

**Discussion**
In conclusion, by elaborately defining the structural geometry for both strong near-field coupling and precise positioning, we demonstrated a simple and flexible approach to integrate FNWs into SWs with a coupling efficiency up to ~97%. The versatility of this approach is experimentally verified by device applications including all-optical modulation and light generation on silicon. Benefitting from the material diversity of FNWs and high reproducibility of the mature top-down fabrication techniques, a variety of novel multifunctional nanophotonic circuits can be expected. For example, by adding metallic[46–49] or organic FNWs[50–52], it is also possible to integrate nanoplasmonics or nanobiology into silicon photonics for diverse opportunities. The SW-assisted micromanipulation we used here is a direct and precise approach to rapid device prototyping and

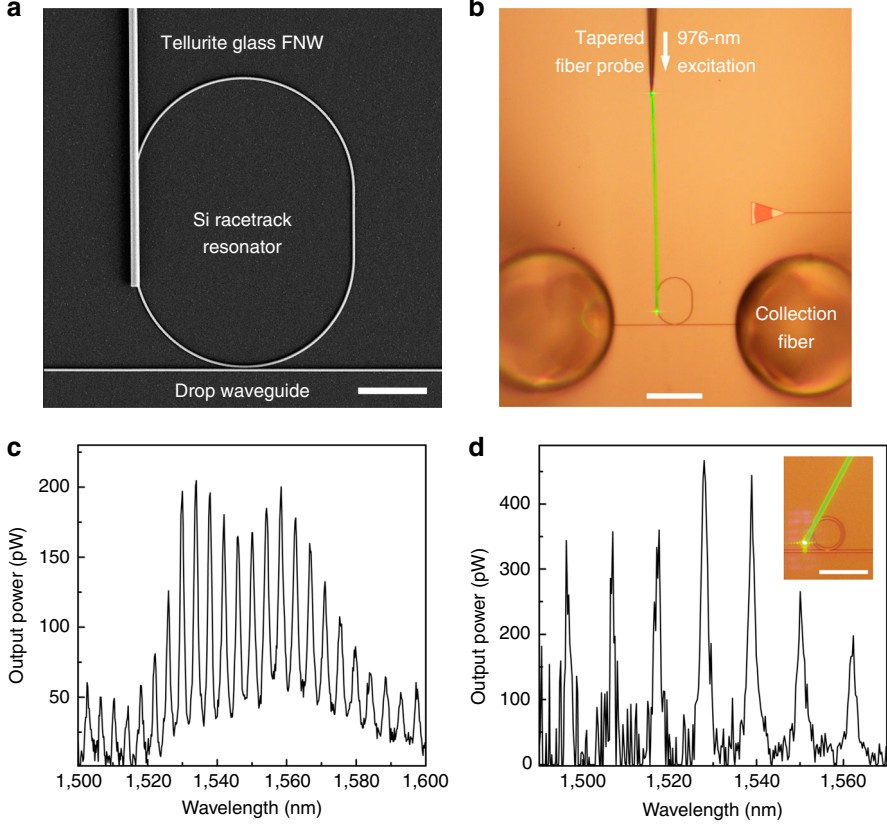

**Fig. 4** Integrated FNW-SW-resonators for on-chip light generation. The active FNWs used here are $Er^{3+}/Yb^{3+}$-codoped tellurite glass nanofibers. **a** SEM image of a 121-µm-perimeter racetrack resonator integrated with a 193-µm-length 980-nm-diameter tellurite glass FNW. Scale bar, 10 µm. **b** Optical micrograph of the integrated FNW-SW-resonators under a 976-nm-wavelength excitation from a tapered fiber probe. Green light observed on the excited FNW is attributed to the up-conversion luminescence of $Er^{3+}$ ions. Scale bar, 50 µm. **c** Output signal collected in the drop waveguide of the racetrack resonator in **b**. **d** Output signal from a 50-µm-perimeter SW resonator integrated with a 960-nm-diameter 216-µm-length tellurite glass FNW. Scale bar, 20 µm

proof-of-concept demonstration. Although at the moment it is still challenging for massive production, our work demonstrated here may open an alternative route to on-chip hybrid integration with high flexibility. Moreover, compared with previous reports on nanophotonic devices assembled with individual nanowires, the reproducibility of the hybrid photonic circuits has been improved largely by the pre-determined coupling length design (Supplementary Note 7). Also, in recent years, rapid progress has been made on deterministic or large-scale nanowire assembly[53–56], which may offer more possibilities to improve the hybrid FNW–SW integration to a scalable technique. Eventually, the flexible integration of nanostructures fabricated by top-down and bottom-up techniques may add a practical pathway to circumventing bottlenecks and continuing the success of both techniques, as well as suggesting a new route to future on-chip nanophotonics.

## Methods

**Fabrication of silicon photonic circuits**. The silicon photonic circuits were fabricated on an SOI wafer with a 340-nm-thick top silicon layer and a 2-µm-thick buried oxide layer. After spin-coating photoresist on the wafer, we used an E-beam lithography process to define silicon-waveguide patterns on the photoresist. Then an inductively coupled plasmon dry-etching process was conducted to fully etch the top silicon layer down to the buried oxide layer with the photoresist mask. For assembling FNWs on the silicon photonic chip with direct contacts, the SWs are air-cladded.

**Synthesis of CdS FNWs**. CdS FNWs were grown by evaporating CdS powder at 800 °C with a high-purity argon flow (150 s.c.c.m.) in a quartz tube furnace. Small silicon substrates coated with Au film were put downstream of the CdS powder to collect the deposited FNWs. After 1.5 h of growth, the tube was cooled down to room temperature and the silicon substrates with as-grown FNWs were taken out for experiments. Scanning electron microscopy and transmission electron microscopy were used for FNW morphology characterization.

**Optical characterization of integrated FNW–SW circuits**. We used an amplified spontaneous emission source for broadband optical characterization of the integrated FNW–SW circuits. Two single-mode fibers (Corning SMF-28) mounted at an angle of 10° were employed to realize input and output coupling with grating couplers of SWs (Fig. 1c). The grating couplers were designed for TE-polarized light with a period of 610 nm and a duty cycle of 50%. Output signals were sent to an optical spectrum analyzer.

**Optical modulation with integrated FNW–SW circuits**. The signal light from a CW 1574.8-nm-wavelength laser was sent into the integrated FNW–SW devices via input gratings. A 405-nm-wavelength CW laser was mechanically chopped to generate switch pulses, which were coupled into a tapered fiber probe to illuminate the target areas of the circuits for optical modulation (Supplementary Note 6). Modulated output was collected by a photodetector and measured by an oscilloscope.

**Data Availability**. The data that support the findings of this study are available from the corresponding author upon request.

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

## Acknowledgements

This work was supported by the National Basic Research Program of China (No. 2013CB328703), the National Natural Science Foundation of China (Nos. 11527901, 61422510), and the Fundamental Research Funds for the Central Universities. We thank Yingxin Xu, Xing Lin, Wei Fang, Shaoliang Yu for helpful discussions. We also thank Prof. Junjie Zhang from China Jiliang University for providing the $Er^{3+}$/$Yb^{3+}$-codoped tellurite glass.

## Author contributions

B.C. and L.T. designed the circuit layouts and the experiment setups. H.W. and D.D. fabricated the SW chips. B.C., H.W., and D.D. performed the numerical calculation. C.X. grew the CdS FNWs. B.C. fabricated the $Er^{3+}$/$Yb^{3+}$-codoped tellurite glass FNWs, assembled the integrated FNW–SW circuits and performed the measurements. B.C., L.T., and D.D. prepared the manuscript. All the authors discussed the results and contributed to the manuscript.

## Additional information

**Competing interests:** The authors declare no competing financial interests.

