## [Peer Review File · Nature Communications]

Reviewers' comments:

Reviewer #1 (Remarks to the Author):

In the work "Flexible integration of free-standing nanowires into silicon photonics" by Chen et al., the authors describe a methodology for integrating free-standing nanowires with silicon photonic optical circuits through micro-manipulation. Furthermore, they demonstrate that it is possible to align the nanowires in such a way as to create high coupling efficiency directional couplers for effective integration. Using these directional couplers, the authors use a number of basic demonstrations to show the efficacy of integrating silicon photonics with free standing wires.

Overall, the manuscript does a good job of both describing the methodology used to manipulate nanowires into silicon photonic circuits, and showing real demonstrations of hybrid devices. While the authors do not discuss the possibility of being able to scale this methodology to make it economically feasible, the results are of interest to the silicon photonics community, especially in the field of all-optical devices, nonlinear devices, and potentially applications such as sensing and integrated lasers.

The reviewer recommends the paper for publication, and offers the following comments to help improve the quality of the manuscript.

***All the images in the paper seem to illustrate a nanowire placed on the top surface of the silicon waveguide (for example, see Fig. 3(a), Fig. 1(b)), however the simulations in the supplementary material show a nanowire side-coupled to the silicon waveguide (for example, see Fig.S5(a)); based on these conflicting depictions, it is not clear what the coupling scheme is. It is suggested that the authors clarify this point. If indeed a vertical coupling scheme is used for the devices in the paper, simulations in the supplementary material should also reflect a vertical coupling scheme.

***If a vertical coupling scheme is used, how tolerant is the coupling efficiency to misalignment. For example, if the nanowire is not perfectly centered with the waveguide, what penalties are expected to be observed

***One of the key points the authors raise in the introduction is low optical loss associated with nanowires due to relatively low surface roughness. However the authors do not comment in the paper about the optical losses characteristic to the wires they fabricate. Have the authors been able to measure the nominal scattering loss the fabricated nanowires?

***The authors state the pulse length and repetition rate in the supplementary material for the all-optical switch, however this information would be interesting to the reader if stated in the manuscript.

***For the MZI created in Fig. 2, it is important to realize a 50/50 split ratio between the CdS nanowire and the silicon waveguide, both at the input and output. Has the split ratio been measured for the coupler used in this device? This could be good information to discuss in the supplementary material.

***The authors show a ring resonator made with a coupled nanowire. It would be of interest to the reader if the Q of the ring was reported.

***The authors show in the supplementary information (Fig.S7) a clever approach for determining the coupling efficiency. This technique is only strictly valid if the nanowire and silicon waveguide share the same propagation loss. If the nanowire has less propagation loss, that savings in loss for OP3 would result in an extracted coupling efficiency that is higher than actuality. It is suggested that the authors discuss this point, and show the impact of the differing propagation loss on this

calculation.

***For the grown nanowires, what are the limitations? Are there limitations on length, or diameter? Are there limitations on nanowire length imparted by the micro-manipulation scheme? This would be important to discuss.

***Are there any pathways for realizing hybrid nanowire/silicon-waveguide devices with electrically active nanowires? For example, would it be possible to do additional fabrication after placing the nanowires to create electrical leads? This would be important for potential applications such as lasers and modulators.

***Are there any pathways for making this type of technique scalable to a large number of devices?

Reviewer #2 (Remarks to the Author):

Report of "Flexible integration of free-standing nanowires into silicon photonics", Bigeng Chen et al.

In this manuscript, the authors present a new platform for integrating free-standing semiconductor nanowires with silicon photonics. They convincingly demonstrate that precise positioning of nanowires in circuits yields highly predictable response with very good performance in terms of coupling losses. The experimental results are of high quality and they cover a range of relevant device geometries.

I believe the work is original and opens up new directions in hybrid photonic circuits. Silicon photonics is a major technological player with many potential applications, however direct integration of active components is still a major bottleneck. This work shows the potential of nanoscale integration through crystalline nanowires as a feasible route toward functional devices. Therefore I am happy to recommend for its publication in Nature Communications

1. In the abstract, it would be better to include the minimum coupling losses in dB as is standard for the integrated photonics field. Close to 100% efficiency would mean losses <0.1 dB?
2. In Figure 1d, they show a strong dependence on the core width. however I would also expect a strong dependence on the gap between the FNW and the waveguide. From Figure 1b and Supplementary figure S3 it is not clear how accurate they can position the FNW in terms of this gap. Can they comment on how this is optimized?
3. On Page 6, they should specify the pulse duration of the 405nm laser used in the photomodulation experiment.
4. Can they comment on the type of modulation used here? Is this thermo-optic or free-carrier induced modulation? It will be good to compare the literature values for the relevant coefficients when stating that CdS has a larger nonlinearity than Si. Also the core of the CdS is larger so this may affect the effective nonlinearity as more field is contained inside the semiconductor.
5. On Page 10, what do they mean with "spectrum-sliced photoluminescence"?
6. In the discussion, they should provide some more scope for the used methodology? What are the prospects of scaling this bottom-up approach? The method currently used appears suitable for prototyping, however a more robust integration strategy may be needed to achieve real-world impact. What would be the most promising application ranges for these types of hybrid devices?

What is the next step in this development?

7. In general, it would help the reader to know what was the yield of the devices. Assuming the ones presented here are 'hero' results, what would be typically the chance of getting such good results when doing this positioning many times? How many devices did they fabricate and what were the results for individual devices that passed certain quality standards?

Response to Reviewer's comments:

Reviewer #1 (Remarks to the Author):

In the work “Flexible integration of free-standing nanowires into silicon photonics” by Chen et al., the authors describe a methodology for integrating free-standing nanowires with silicon photonic optical circuits through micro-manipulation. Furthermore, they demonstrate that it is possible to align the nanowires in such a way as to create high coupling efficiency directional couplers for effective integration. Using these directional couplers, the authors use a number of basic demonstrations to show the efficacy of integrating silicon photonics with free standing wires.

Overall, the manuscript does a good job of both describing the methodology used to manipulate nanowires into silicon photonic circuits, and showing real demonstrations of hybrid devices. While the authors do not discuss the possibility of being able to scale this methodology to make it economically feasible, the results are of interest to the silicon photonics community, especially in the field of all-optical devices, nonlinear devices, and potentially applications such as sensing and integrated lasers.

The reviewer recommends the paper for publication, and offers the following comments to help improve the quality of the manuscript.

1. All the images in the paper seem to illustrate a nanowire placed on the top surface of the silicon waveguide (for example, see Fig. 3(a), Fig. 1(b)), however the simulations in the supplementary material show a nanowire side-coupled to the silicon waveguide (for example, see Fig.S5(a)); based on these conflicting depictions, it is not clear what the coupling scheme is. It is suggested that the authors clarify this point. If indeed a vertical coupling scheme is used for the devices in the paper, simulations in the supplementary material should also reflect a vertical coupling scheme.

Response: Thank you for mentioning this point. We are sorry for the confusion about the side-/vertical-coupling methods. For the side-coupling scheme, the nanowire was supported by the SiO₂ substrate instead of the silicon core, as shown in Fig. 1 and Fig. 2. In this case, when assembling the nanowire-waveguide coupling devices, we placed a nanowire on the SiO₂ substrate and then pushed it towards a target silicon waveguide (Supplementary Fig. 3) until it was stopped by the silicon core. We have also double-checked the side-coupling structure by using a top-viewed SEM image depicted in the lower panel of Fig. 1b. From this figure, one sees the nanowire and silicon waveguide simultaneously, which means that the nanowire is at the *side* of the silicon core. In contrast, for the vertical coupling case, a part of the silicon waveguide can not be seen because the nanowire on the *top* is larger than the silicon core, as

shown in Figs. 3b-3c and 4a. We have clarified this point by adding a statement into Section 2 of the supplementary material.

2. If a vertical coupling scheme is used, how tolerant is the coupling efficiency to misalignment. For example, if the nanowire is not perfectly centered with the waveguide, what penalties are expected to be observed

Response: Thank you for raising this important point. For the center misalignment in vertical coupling scheme, we used Lumerical FDTD to explore its tolerance and penalties. As shown below, when an 860-nm-diameter CdS nanowire and a 290-nm-width silicon waveguide are vertically coupled with a coupling length of 3 μm , centre misalignment up to 200 nm only cause 0.4 dB reduction in coupling efficiency. The robustness can be attributed to the strong evanescent field overlap of the nanowire and waveguide. This result has been added into the Section 4 of the supplementary material.

Figure R1 Coupling efficiencies from a 290-nm-width silicon waveguide to an 860-nm-diameter CdS nanowire of different centre misalignments with a vertical coupling scheme.

3. One of the key points the authors raise in the introduction is low optical loss associated with nanowires due to relatively low surface roughness. However the authors do not comment in the paper about the optical losses characteristic to the wires they fabricate. Have the authors been able to measure the nominal scattering loss the fabricated nanowires?

Response: Thank you for mentioning this key point. Previous reports had shown that nanowire propagation loss coefficient can be obtained by measuring the propagation-distance-dependent output intensity (Y. Ma et al. *Opt. Lett.* 2010, **35**, 1160–1162; G. Qu et al. *Adv. Funct. Mater.* 2013, **23**, 1232–1237). However, the validity of this method is restricted to the nanowires with obvious propagation losses. The light coupled from a tapered fiber into a nanowire will firstly suffer from significant leaky-mode loss before the supported mode become stable. If the nanowire cannot provide obvious propagation loss with sufficient propagation distance, the

measured loss will be dominated by the initial leaky-mode loss, not to mention its reliability limited by the detection precision. On the other hand, according to our measured result shown in Fig. 1d, the propagation loss of the coupled nanowire should be at least very close to the silicon waveguide's, otherwise the actual coupling efficiency would be over unity as we obtained near 100% measured efficiency at around 1,585 nm. As the propagation loss coefficient of the silicon waveguides is about 2-5 dB/cm at telecom wavelengths (S. Chen et al. *J. Lightwave Technol.* 2015, **33**, 2279–2285) and the nanowires have ultralow RMS surface roughness, we believe that the nanowire's loss coefficient is below that level (i.e., <2-5 dB/cm). Therefore to measure the exact propagation loss, we need a nanowire with a length of several millimetres, which is currently beyond what we can obtain from our grown samples (100- μ m level). Because the proper growth technique for long nanowires is still under investigation that will probably cost a long time, we are sorry for being not able to present a reliable nanowire propagation loss data here. However the coupling efficiency measurement result and the RMS surface roughness of 0.3 nm (see Supplementary Fig. 1c, approaching the similar level of silica nanofiber) imply that the scattering loss coefficient should be much lower than 1 dB/cm. This discussion of propagation loss estimation for the nanowire has been added into Section 5 of the supplementary material.

4. The authors state the pulse length and repetition rate in the supplementary material for the all-optical switch, however this information would be interesting to the reader if stated in the manuscript.

Response: Thank you for pointing out this issue. We have added the 0.75-ms pulse length and the 67-Hz repetition rate of the switch pulses in the manuscript.

5. For the MZI created in Fig. 2, it is important to realize a 50/50 split ratio between the CdS nanowire and the silicon waveguide, both at the input and output. Has the split ratio been measured for the coupler used in this device? This could be good information to discuss in the supplementary material.

Response: Thank you for mentioning this point. We haven't measure the split ratio but we managed to obtain the calculated values by simulating the nanowire-waveguide coupler with the same structural parameters used in the experiment. As shown below, we can see the coupling efficiencies from 1550 nm to 1620 nm are close to 50% and the largest split ratio is only 57/43. In addition, the measured transmission spectrum of the MZI (Fig. 2b) shows high transmissions (0.46 dB in average) at the constructive interference positions as well as large extinction ratios, which also suggest that the split ratios at both the input and output couplers were very close to 50/50. This discussion has been added into the Section 6 of the supplementary material.

Figure R2 Calculated transmissions from the outputs of silicon waveguide and CdS nanowire in the couplers used for the MZI.

6. The authors show a ring resonator made with a coupled nanowire. It would be of interest to the reader if the Q of the ring was reported.

Response: Thank you for this suggestion. According to the measured spectrum in Fig. 3d, the Q factor of the ring resonator is around 1400. This information is already in the section “Integrated FNW-SW racetrack resonator using a vertical coupling scheme” of the manuscript.

7. The authors show in the supplementary information (Fig.S7) a clever approach for determining the coupling efficiency. This technique is only strictly valid if the nanowire and silicon waveguide share the same propagation loss. If the nanowire has less propagation loss, that savings in loss for OP3 would result in an extracted coupling efficiency that is higher than actuality. It is suggested that the authors discuss this point, and show the impact of the differing propagation loss on this calculation.

Response: Thank you for mentioning this point. When the nanowire and waveguide have different losses, the actual efficiency becomes $(OP3 - loss_{FNW} - OP1 + loss_{SW})/2$, which means for $(OP3 - OP1)/2$ there is an error of $(loss_{SW} - loss_{FNW})/2$. However, as the loss coefficient of the SWs we used here was about 2-5 dB/cm (S. Chen et al. *J. Lightwave Technol.* 2015, **33**, 2279–2285), the 26- μ m-length straight SW only introduced a loss of 0.005-0.013 dB. Therefore, considering the largest loss of the SW, this error becomes $(0.013 \text{ dB} - loss_{FNW})/2$, and is only ~ 0.007 dB in maximum provided $loss_{FNW}$ is negligible. Discussion of this point has been added into Section 5 of the supplementary information. On the other hand, by zooming in the spectra of the measured coupling efficiencies shown in Fig.1d, we obtained the maximum coupling efficiency as -0.13 dB at around 1585 nm. Considering the possible error of 0.007 dB, the actual maximum coupling efficiency should be no less than -0.14 dB (97%). We have also corrected the coupling efficiency from $\sim 100\%$ to 97% in the manuscript.

8. For the grown nanowires, what are the limitations? Are there limitations on length, or diameter? Are there limitations on nanowire length imparted by the micro-manipulation scheme? This would be important to discuss.

Response: Thank you for raising these questions. Precise control over length, diameter, growth direction, morphology and composition had been realized for the growth of nanowires. For example, the length can be controlled from several to hundreds of micrometres, while the diameter from several nanometres to hundreds of nanometres can also be determined by the size of the metal alloy droplet catalyst (R. Yan et al. *Nature Photon.* 2009, **3**, 569–576). Generally, longer and thicker nanowires are easier to be manipulated. However, using the nanoscale fibre tip shown in Supplementary Fig. 2a, we are also able to manipulate very tiny nanowires with a length down to 8 μm (J. Li et al. *Adv. Mater.* 2013, **25**, 833–837) and/or a diameter down to 100 nm (X. Wu et al. *Nano. Lett.* 2013, **13**, 5654–5659). We believe one major limitation of the grown nanowires is the difficulty of accurate deterministic positioning for device assembling in large scale (M. Kwiat et al. *Nano Today* 2013, **8**, 677–694). This part of discussion has been added into Section 2 of the supplementary material.

9. Are there any pathways for realizing hybrid nanowire/silicon-waveguide devices with electrically active nanowires? For example, would it be possible to do additional fabrication after placing the nanowires to create electrical leads? This would be important for potential applications such as lasers and modulators.

Response: Thank you for mentioning this point. Actually the nanowires are very stable once they are totally attached on a clean substrate. For example, after we spin-coated a PMMA layer on a SiO₂ substrate with dispersed nanowires at a speed of 3000 rpm, the nanowires stayed still as before (shown below). This property provides the feasibility of subsequent top-down processing after nanowires are in positions with silicon waveguides. Actually there had been several literature reports on nanowire-based photonic devices with fabricated electrical leads (e.g., C. J. Barrelet et al. *Nano. Lett.* 2004, **4**, 1981–1985; F. Qian et al. *Nano. Lett.* 2005, **5**, 2287–2291). Therefore it is probable to realize hybrid nanowire/silicon-waveguide devices with electrically active nanowires by adding electrical leads. A statement for this point has been added into Section 2 of the supplementary material.

Figure R3 Optical micrographs of dispersed CdS nanowires on a SiO₂ substrate before and after spin-coating a PMMA layer.

10. Are there any pathways for making this type of technique scalable to a large number of devices?

Response: Thank you for pointing out this issue. As we mentioned above, it's still challenging to assemble nanowire devices accurately in large scale, especially for nanophotonic application. However, recently there have been a lot of reports on deterministic assembling of grown nanowires by different approaches such as nanowire guided growth (M. Schwartzman et al. *Proc. Natl Acad. Sci. USA* 2013, **110**, 15195–15200; X. Miao et al. *Nano Lett.* 2015, **15**, 2780–2786) and nanoscale combing technique (J. Yao et al. *Nature Nanotech.* 2013, **8**, 329–335) and shape-controlled assembling (Y. Zhao et al. *Nano Lett.* 2016, **16**, 2644–2650; Y. S. No et al. *ACS Photon.* 2016, DOI-10.1021/acsphotonics.6b00775). These progresses imply that the coupling of nanowires and silicon waveguides is also possible to be scalable to a large number of devices. This part of discussion has been added into the discussion section of the manuscript.

Reviewer #2 (Remarks to the Author):

Report of "Flexible integration of free-standing nanowires into silicon photonics", Bigeng Chen et al.

In this manuscript, the authors present a new platform for integrating free-standing semiconductor nanowires with silicon photonics. They convincingly demonstrate that precise positioning of nanowires in circuits yields highly predictable response with very good performance in terms of coupling losses. The experimental results are of high quality and they cover a range of relevant device geometries.

I believe the work is original and opens up new directions in hybrid photonic circuits. Silicon photonics is a major technological player with many potential applications, however direct integration of active components is still a major bottleneck. This work shows the potential of nanoscale integration through crystalline nanowires as a feasible route toward functional devices. Therefore I am happy to recommend for its publication in Nature Communications

1. In the abstract, it would be better to include the minimum coupling losses in dB as is standard for the integrated photonics field. Close to 100% efficiency would mean losses <0.1 dB?

Response: Thank you for pointing out this issue. After considering the measurement error for the coupling efficiency mentioned in comment #7 from reviewer #1, we determine a precise efficiency of 97% instead of the approximate value of $\sim 100\%$, which means the losses is around 0.14 dB. Corresponding statement has been added in the abstract.

2. In Figure 1d, they show a strong dependence on the core width. However I would also expect a strong dependence on the gap between the FNW and the waveguide. From Figure 1b and Supplementary figure S3 it is not clear how accurate they can position the FNW in terms of this gap. Can they comment on how this is optimized?

Response: Thank you for mentioning this point. Actually, when assembling the nanowire-waveguide coupling devices, we just pushed the nanowire towards a target waveguide as shown in Supplementary Figure 3 until it was stopped by the waveguide. It means they were in contact finally and the gap between them should be almost zero as designed. On the other hand, we noticed that the sidewall roughness of the silicon waveguide is less than 10 nm typically. As a result, the nanowire-waveguide contact might not be perfect and there is a tiny gap. Therefore we have also checked the dependence of the coupling efficiency on the gap by using an FDTD simulation. As shown below, for both coupling directions, when the gap increases from 5 nm to 15 nm, the maximum efficiency decreases by about 0.3 dB and the spectrum blueshifts by about 5 nm. We have added these simulation results into Section 4 of the supplementary material. Considering the nanowire's surface RMS roughness (<0.5 nm) and the side-wall roughness peak amplitude of silicon waveguides (<5 nm, see e.g., Y. A. Vlasov et al. *Opt. Express* 2004, **12**, 1622–1631), the possible gaps between the nanowires and waveguides in our experiment should be around or less than 5 nm and have very small effect on coupling efficiency. We have also calculated the coupling efficiencies for Supplementary Fig. 6 by adding a 5-nm gap, which shows very small differences compared with the previous result.

Figure R4 Coupling efficiencies between an 860-nm-diameter CdS nanowire and a 290-nm-width silicon waveguide with different gaps for both coupling directions.

3. On Page 6, they should specify the pulse duration of the 405nm laser used in the photomodulation experiment.

Response: Thank you for the nice suggestion. The 0.75-ms pulse duration of the 405-nm laser has been added into the relevant part on Page 6.

4. Can they comment on the type of modulation used here? Is this thermo-optic or free-carrier induced modulation? It will be good to compare the literature values for the relevant coefficients when stating that CdS has a larger nonlinearity than Si. Also the core of the CdS is larger so this may affect the effective nonlinearity as more field is contained inside the semiconductor.

Response: Thank you for providing these comments. Take the silicon waveguide as an example, according to the signal light wavelength position shown in Supplementary Figure 10c and the positive modulation direction shown in Figure 2d, we deduced that its refractive index decreased when illuminated by the 405nm laser, which is consistent with the free-carrier induced nonlinear refraction of silicon (J. Leuthold et al. *Nature Photon.* 2010, **4**, 535–544). If the thermal-optic effect dominated, the refractive index should increase as silicon's thermal nonlinear coefficient is positive (G. Cocorullo et al. *Opt. Lett.* 1994, **19**, 420–422). The situation is also similar for the CdS nanowire (N. Venkatram et al. *J. Appl. Phys.* 2006, **100**, 074309). Therefore we believe the modulation type used here is free-carrier induced modulation. Relevant statement and discussion have been added into the manuscript and Section 6 of the supplementary material, respectively.

For silicon at 1550-nm wavelength, the carrier-induced refractive index change can be expressed as $\Delta n = \Delta n_e + \Delta n_h = -(8.8 \times 10^{-22} N_e / \text{cm}^{-3} + 8.5 \times 10^{-18} (N_h / \text{cm}^{-3})^{0.8})$, where N_e and N_h are the electron density and hole density respectively (G. T. Reed et al. *Nature Photon.* 2010, **4**, 518–526). However, to the best of our knowledge, so far there is no report regarding this characteristic of CdS at telecom wavelengths yet. We apologize for making the

misunderstanding by stating a larger carrier-induced nonlinearity of CdS that sounds like it had been studied in the same wavelength range as silicon before. Actually this is a preliminary conclusion that could be drawn from the stronger modulation effect of the CdS nanowire compared with the silicon waveguide (Figure 2d). We have revised this statement in the manuscript. As the higher photoconductivity of CdS mentioned in Ref.40 (T. Y. Wei et al. *Appl. Phys. Lett.* 2010, **96**, 013508) cannot guarantee it has larger nonlinearity (although it does show a larger nonlinearity in this work), we have removed this reference from our revised manuscript.

Because the free-carrier induced nonlinearity was used, it was the photo-excited carrier density that determined how much the refractive index can be changed. Accordingly, we should consider the field intensity (or power density) interacting with the silicon waveguide or CdS nanowire. For example, when the 300-nm-diameter waveguide and 860-nm-diameter nanowire were illuminated by a Gaussian beam with a spot size of 6 μm respectively, the field intensities interacting with them were almost identical as the illuminations along their width direction were nearly uniform and the illuminated lengths were also the same. Therefore the core sizes of the waveguide and nanowire is less critical here for nonlinearity comparison.

5. On Page 10, what do they mean with "spectrum-sliced photoluminescence"?

Response: Thank you for mentioning this point. The term "spectrum-sliced" was raised in some previous reports where arrayed waveguide gratings were used to filter out multiple lightwaves with discrete wavelengths from broadband light sources (J. S. Lee et al. *IEEE Photon. Technol. Lett.* 1993, **5**, 1458-1461; Y. Takushima et al. *IEEE Photon. Technol. Lett.* 1999, **11**, 322-324). Here we also used the similar concept by filtering the broadband photoluminescence from the $\text{Er}^{3+}/\text{Yb}^{3+}$ -codoped nanowires with the silicon waveguide resonators. Relevant references have been added in the manuscript.

6. In the discussion, they should provide some more scope for the used methodology? What are the prospects of scaling this bottom-up approach? The method currently used appears suitable for prototyping, however a more robust integration strategy may be needed to achieve real-world impact. What would be the most promising application ranges for these types of hybrid devices? What is the next step in this development?

Response: Thank you for pointing out this issue. The micromanipulation we used here is an easy method for rapid device prototyping and new concept demonstration without the requirement of complicated facilities and process flows. Although at the moment it is still challenging to be developed into a massive manufacturing technique, our work proposed here could be the initial step toward a new scheme for on-chip hybrid integration devices with large-

scale fabrication ability. Compared with previous reports on nanophotonic devices assembled with individual nanowires, the reproducibility of the hybrid photonic circuits has been improved largely by the pre-determined coupling length design. Besides, there have been a lot of studies on deterministic assembling of grown nanowires with different approaches such as nanowire guided growth (M. Schwartzman et al. *Proc. Natl Acad. Sci. USA* 2013, **110**, 15195–15200; X. Miao et al. *Nano Lett.* 2015, **15**, 2780–2786), nanoscale combing technique (J. Yao et al. *Nature Nanotech.* 2013, **8**, 329–335) and shape-controlled assembling (Y. Zhao et al. *Nano Lett.* 2016, **16**, 2644–2650; Y. S. No et al. *ACS Photon.* 2016, DOI-10.1021/acsphotonics.6b00775). These progresses imply that the integration of nanowires and silicon waveguides is possible to be scalable to achieve real-world impact.

With the great material diversity of nanowires, we can have a large number of candidates for many on-chip applications such as light generation, modulation and detection with higher performances, supplementing the material base in the conventional top-down technique. Therefore we believe on-chip optical communication would be one of the most promising applications for the hybrid devices in the near future. Next step we can start a systematic research on robust large-scale integration strategy, as well as integrating other nanostructures into silicon photonics such as nanorods, nanosheets, plasmonic nanowires and so on, to explore more potential applications of this hybrid integration.

This part of discussion has been added into the discussion section of the manuscript.

7. In general, it would help the reader to know what was the yield of the devices. Assuming the ones presented here are 'hero' results, what would be typically the chance of getting such good results when doing this positioning many times? How many devices did they fabricate and what were the results for individual devices that passed certain quality standards?

Response: Thank you for mentioning this point. The key factor of the integration experiment is the phase matching between the nanowires and silicon waveguides, which means the core sizes of these two structures and the coupling lengths should be carefully chosen to realize proper coupling efficiency. With the aid of numerical simulation, we can determine a certain range of the structural parameters. For a person skilled in micromanipulation, the nanowire positioning is not very difficult and thus appears less critical. For example, in the side-by-side coupling scheme, we just needed to push the nanowires to contact with the target silicon waveguide as the coupling lengths had been pre-determined by the straight section after the S-bend (Figure 1b). In detail, 20 couplers were fabricated for coupling efficiency measurement, among which there were 4 showing over 80% coupling efficiencies. For the MZIs, due to further structural optimization, half of the 20 fabricated devices showed large extinction ratio over 10 dB. As to hybrid resonators with vertical coupling scheme, the positioning was a bit more challenging because the nanowire was to be suspended across two waveguides collinearly.

Among the 15 hybrid resonators, there were 4 with Q factors over 1000. For the resonators integrated with Er³⁺/Yb³⁺-codoped nanowires, half of the 10 devices gave maximum optical output over 200 pW. Generally, we believe better optimization in design can help reduce the difficulty in experiment. This discussion has been added into Section 7 of the supplementary material.

List of changes

Main text manuscript:

1. In the whole manuscript, the words “Figure” are replaced by “Fig.” except at starts of sentences. In “References”, we have removed 1 paper (previous Ref. 40) and added 4 papers (Ref. 53-56). The reference orders are renumbered. An additional affiliation of Limin Tong, previously lost, was added as “Collaborative Innovation Center of Extreme Optics, Shanxi University, Taiyuan 030006, China”.
2. In the 9th line of the 1st paragraph (Abstract), the word “100%” is changed to “97% (i.e. ~0.14 dB loss)”.
3. In the 4th line of the 3rd paragraph, the word “100%” is changed to “97%”; in the 5th line, the words “corresponding to a minimum loss of ~0.14 dB” are added; in the 6th line, the word “were” is changed to “are”.
4. In the 8th line of the 4th paragraph (Section “Optical near-field coupling between FNWs and SWs”), the words “Figure 1b” are replaced by “Figs. 1b and 1c”; in the 19th line the words “with nearly 100% around 1,585 nm” are removed, and the words “with a maximum value of about 97% around 1,585 nm after calibrating the propagation losses of the FNW and the SW (Supplementary Section 5)” is added.
5. In the 7th line of the 5th paragraph (Section “Integrated FNW-SW MZI for all-optical modulation”), the sentence “Compared with silicon, CdS has higher nonlinearity induced by photo-excited carriers under similar optical excitation owing to its higher photoconductivity⁴⁰, and the integration of CdS FNW can greatly enhance the nonlinear response of an MZI circuit” is removed; in the 10th line, the words “To generate free carriers for refractive index modulation similar to silicon optical modulators⁹” are added; in the 11th line, the word “chopped” is added; in the 12th line, the words “repetition rate ~67 Hz, pulse duration ~0.75 ms, and” are added; in the 17th line, the reference number “41” is changed to “40”; in the 18th line, the word “comparison” is changed to “reference”; in the 19th line, the word “became” was changed to “was”; in the 20th line, the sentence “The results show that, compared with SW, CdS FNW presented higher carrier-induced optical nonlinearity, and thus offered higher modulation depth in the integrated circuit.” are added; in the 22th line, the reference number “42” is changed to “41”; in the 23th line, the word “also” was removed. Figure 2 is replaced by the figure shown below, where the schematic of the MZI in Fig. 2c is modified to depict the S-bend waveguides for quasi-self-aligned assembling, and the words “U shape Si waveguide” in Fig. 2a are changed to “U-shape SW”.

6. In the Fig. 3c the words “Bus waveguide” are moved to the right side of the bus waveguide and rotated by 90° as shown below:

7. In the 3rd line of the 7th paragraph (Section “**Integrated FNW-SW resonators for on-chip light generation**”), the word “**vertically**” is added; in 5th line, the reference numbers “42, 43” are added; in the 7th line, the reference number “43” is changed to “44”; in the 8th line, the reference number “44” is changed to “45”.

8. In the 3rd line of the 8th paragraph (Section “**Discussion**”), the word “100%” is changed to “97%”; in the 7th line, the reference numbers “45-48” and “49-51” are changed to “46-49” and “50-52”, respectively; in the 8th line, a statement is added as “**The SW-assisted micromanipulation we used here is a direct and precise approach to rapid device prototyping and new concept demonstration. Although at the moment it is still challenging for massive production, our work demonstrated here may open a new route to on-chip hybrid integration with high flexibility. Moreover, compared with previous reports on nanophotonic devices assembled with individual nanowires, the reproducibility of the hybrid photonic circuits is largely improved by the pre-determined coupling length design (Supplementary Section 7). Also, in recent years, rapid progresses have been made on deterministic or large-scale nanowire assembly⁵³⁻⁵⁶, which may offer more possibilities to improve the hybrid FNW-SW integration to a scalable technique**”.

9. In the 4th line of the subsection “**Optical modulation with integrated FNW-SW circuits**” in “**Methods**”, the sentence “**The repetition and pulse duration of the switch light were about 67 Hz and 0.75 ms, respectively**” is removed.

10. In the 2nd line of the “**Acknowledge**”, the words “No. 11527901” are changed to “**Nos. 11527901, 61422510**”.

11. The data availability statement is added as “The data that support the findings of this study are available from the corresponding author upon request” at the end of “**Methods**”.

Supplementary Information:

1. In the whole manuscript, the words “Figure” are replaced by “**Fig.**” except at starts of sentences. For the supplementary figures, the words “**Supplementary**” are added before the words “Fig.” and the letters “S” before the figure numbers are removed. In “**References**”, we have removed 1 paper (previous **Ref. 8**) and added 7 papers (**Ref. 3, 4, 7, 8, 9, 11, 12**). The reference orders are renumbered. An additional affiliation of Limin Tong, previously lost, was added as “**Collaborative Innovation Center of Extreme Optics, Shanxi University, Taiyuan 030006, China**”.

2. In the 10th line of the 1st paragraph of Section 2 “**Micromanipulation of FNWs for integration with silicon waveguides (SWs)**”, a statement “**Generally, longer and thicker FNWs are easier to be manipulated. However, using the nanoscale fibre tip shown in Supplementary Fig. 2a, we are also able to**

manipulate tiny FNWs with a length down to 8 μm [Ref. 3] and/or a diameter down to 100 nm [Ref. 4]] is added; Supplementary Fig. 2 is moved from after the 2nd paragraph to before the 2nd paragraph together with its caption; in the 8th line of the 2nd paragraph, a sentence “Being totally attached on a clean substrate, FNWs are usually stable enough for afterward top-down processing, which suggests that electrically active FNWs can also be used for this FNW-SW integration by adding electrical leads” is added.;

The 3rd paragraph is added as “For the side-coupling scheme, the nanowire was supported by the SiO₂ substrate instead of the silicon core, as shown in Fig. 1 and Fig. 2 in the main text. In this case, when assembling the nanowire-waveguide coupling devices, we placed a nanowire on the SiO₂ substrate and then pushed it towards a target silicon waveguide (Supplementary Fig. 3) until it was stopped by the silicon core. We double-checked the side-coupling structure by using a top-viewed SEM image depicted in the lower panel of Fig. 1b. From this figure, one sees the nanowire and silicon waveguide simultaneously, which means that the nanowire is at the *side* of the silicon core. In contrast, for the vertical coupling case, a part of the silicon waveguide can not be seen because the nanowire on the *top* is larger than the silicon core, as shown in Figs. 3b-3c and 4a”.

2. In the 4th line of Section 3 “Effective indices of CdS FNWs and SWs”, the reference numbers “3” and “4” are changed to “5” and “6”, respectively.

3. In the 2nd line of the 1st paragraph of Section 4 “Simulation of the near-field optical coupling between a CdS FNW and an SW”, the word “side-by-side” is added; in the caption for Supplementary Fig. 5a, the word “side-by-side” is added.

In the 1st line of the 2nd paragraph, the words “with the side-by-side coupling scheme” is added, followed by a new sentence of “Considering the nanowire’s surface RMS roughness (<0.5 nm) and the side-wall roughness peak amplitude of silicon waveguides⁷ (<5 nm), the gap between the FNW and the SW was set as 5 nm”; in the 8th line the words “of the efficiency dependence on SW width (shift by about 10 nm)” are added; in the 10th line the words “and the difference between the actual CdS refractive index and literature value⁶ at telecom wavelengths” are added; Supplementary Fig. 6 is replaced by a new graph as below:

The 3rd paragraph is added as “We checked the dependence of the coupling efficiency on the gap by using an FDTD simulation. Supplementary Fig. 7 gives the calculated coupling efficiencies between an 860-nm-diameter CdS FNW and a 290-nm-width SW with 3 gap sizes. As shown below, for both coupling directions, when the gap increases from 5 nm to 15 nm, the maximum efficiency decreases by about 0.3 dB and the spectrum buleshifts by about 5 nm”, together with Supplementary Fig. 7 and the corresponding caption as shown below:

Supplementary Figure 7. Coupling efficiencies between an 860-nm-diameter CdS FNW and a 290-nm-width SW with different gaps for both coupling directions.

The 4th paragraph is added as “We also calculated the coupling efficiencies between a CdS FNW and a SW in a vertical coupling scheme regarding different centre misalignments. As shown in Supplementary Fig. 8, when an 860-nm-diameter CdS FNW and a 290-nm-width SW are vertically coupled with a coupling length of 3 μm , misalignment up to 200 nm only causes 0.4-dB reduction in coupling efficiency. The robustness can be attributed to the strong evanescent field overlap of the FNW and the SW”, together with Supplementary Fig. 8 and the corresponding caption as shown below:

Supplementary Figure 8. Coupling efficiencies from an 860-nm-diameter CdS FNW to a 290-nm-width SW in a vertical coupling scheme with different centre misalignments

4. In the 2nd line of the 1st paragraph of Section 5 “**An experimental approach to obtaining coupling efficiency between a CdS FNW and an SW**”, the figure numbers “S7a-S7c” are changed to “9a-9c”; in the 2nd paragraph, the word “it” is added in the 1st line, and the figure numbers “S7e” in the 2nd line, “S7b” in the 3rd line, “S7e” in the 4th line, “S7c” in the 5th line, “S7e” in the 6th line, “S7d” in the 8th line are changed to “9e”, “9b”, “9e”, “9c”, “9e” and “9d”, respectively.

The 3rd paragraph is added as “In principle, this approach is valid only when the CdS FNW and the SW share a same propagation loss. When they have different losses, the actual efficiency becomes $(OP3 - loss_{FNW} - OP1 + loss_{SW})/2$, which means for $(OP3 - OP1)/2$ there is an error of $(loss_{SW} - loss_{FNW})/2$. However, as the loss coefficient of the SWs was 2-5 dB/cm [Ref. 8], the 26- μ m-length straight SW only introduced a loss of 0.005-0.013 dB. Therefore, considering the largest loss of the SW, this error becomes $(0.013 \text{ dB} - loss_{FNW})/2$, and is only 0.007 dB in maximum provided $loss_{FNW}$ is negligible. On the other hand, by zooming in the spectra of the measured coupling efficiencies shown in Fig. 1d, we obtained the maximum coupling efficiency as -0.13 dB at around 1,585 nm. Considering the possible error of 0.007 dB, the actual maximum coupling efficiency should be no less than -0.14 dB (97%), which is very close to the -0.12 dB obtained from Supplementary Fig. 6”.

5. In Section 6, the heading of “**Wavelength selection for signal light in the optical modulation**” is changed to “**Optical modulation with the hybrid MZI**”; in the 1st paragraph, the figure numbers “S8a” and “S8b” in the 4th line are changed to “10a” and “10b” respectively, the reference numbers “[Ref. 5]” and “[Ref. 6]” in the 5th line are changed to superscript numbers “⁹” and “¹⁰” respectively; in the 2nd paragraph, the figure numbers “S8c” in the 1st line and the 3rd line are both changed to “10c”, the figure number “10” for the caption is also changed from “S8”; A sentence “Thermal nonlinear refraction effect was believed to play a minor role here, otherwise the signs of the thermal nonlinear coefficients of CdS¹¹ and silicon¹² would produce opposite modulation according to the signal light position in Supplementary Fig. 10c” is added in the last of the 2nd paragraph.

The 3rd paragraph is added as “For an MZI, it is important to realize 50/50 split ratios at both the input and output couplers to achieve large extinction ratios. As a supplementary investigation to the experiment, numerical calculations (Supplementary Fig. 11) for the transmissions of the couplers were also performed with the same structural parameters as in the experiment, including the bended coupling SW formed by the connected ends of two S-bends. We can see the coupling efficiencies from 1550 nm to 1620 nm are close to 50% and the largest split ratio is only 57/43. In addition, from the measured transmission spectrum of the MZI (Fig. 2b) high transmissions (0.46 dB in average) at the constructive interference positions were also observed as well as the large extinction ratios. These results suggest that the split ratios at both the input and output couplers were very close to 50/50”, together with the Supplementary Fig. 11 and its caption as shown below.

Supplementary Figure 11. Calculated transmissions from the outputs of SW and CdS FNW in the couplers used for the MZI.

The 4th paragraph is added as “When the 300-nm-diameter waveguide and 860-nm-diameter nanowire were illuminated by a Gaussian beam with a spot size of 6 μm respectively, the field intensities interacting with them were almost identical as the illuminations along their width direction were nearly uniform and the illuminated lengths were also the same. Therefore the core sizes of the waveguide and nanowire is less critical here for nonlinearity comparison”.

6. Section 7 “**Reproducibility of the integrated FNW-SW device fabrication**” is added as “Before the micromanipulation with the FNWs, we used numerical simulation to determine a certain range of the structural parameters for fabrication. 20 couplers were fabricated for coupling efficiency measurement, among which there were 4 showing over 80% coupling efficiencies. For the MZIs, due to further structural optimization, half of the 20 fabricated devices showed large extinction ratio over 10 dB. As to hybrid resonators with vertical coupling scheme, the assembling was a bit more challenging because the FNW was to be suspended across two SWs collinearly. Among the 15 hybrid resonators, there were 4 with Q factors over 1000. For the resonators integrated with $\text{Er}^{3+}/\text{Yb}^{3+}$ -codoped FNWs, half of the 10 devices gave maximum optical output over 200 pW. Generally, we believe better optimization in design can help reduce the difficulty in experiment”.

REVIEWERS' COMMENTS:

Reviewer #2 (Remarks to the Author):

I congratulate the authors with the very nice work. They have been able to respond to all comments to satisfaction. I am therefore happy to recommend for publication of the article in Nature Communications.